# Modified WCRF/AICR Score and All-Cause, Digestive System, Cardiovascular, Cancer and Other-Cause-Related Mortality: A Competing Risk Analysis of Two Cohort Studies Conducted in Southern Italy

**DOI:** 10.3390/nu13114002

**Published:** 2021-11-10

**Authors:** Antonella Mirizzi, Laura R. Aballay, Giovanni Misciagna, Maria G. Caruso, Caterina Bonfiglio, Paolo Sorino, Antonella Bianco, Angelo Campanella, Isabella Franco, Ritanna Curci, Filippo Procino, Anna M. Cisternino, Maria Notarnicola, Pierina F. D’Aprile, Alberto R. Osella

**Affiliations:** 1Laboratory of Epidemiology and Biostatistics, National Gastroenterology Institute, “S. de Bellis” Research Hospital, Via Turi 27, 70013 Castellana Grotte, Bari, Italy; dottoressamirizzi@yahoo.com (A.M.); catia.bonfiglio@irccsdebellis.it (C.B.); paolo.sorino@irccsdebellis.it (P.S.); antonella.bianco@irccsdebellis.it (A.B.); angelo.cmpanella@irccsdebellis.it (A.C.); isabella.franco@irccsdebellis.it (I.F.); ritanna.curci@irccsdebellis.it (R.C.); filippoprocino@yahoo.it (F.P.); francesca.daprile@irccsdebellis.it (P.F.D.); 2Human Nutrition Research Center (CenINH), School of Nutrition, Faculty of Medical Sciences, Universidad Nacional de Córdoba, Córdoba X5000, Argentina; laballay@fcm.unc.edu.ar; 3Scientific and Ethical Committee Polyclinic Hospital, University of Bari, 70124 Bari, Puglia, Italy; gmisciag@libero.it (G.M.); gabriella.caruso@irccsdebellis.it (M.G.C.); 4Clinical Nutrition Outpatients Clinic, National Gastroenterology Institute, “S. de Bellis” Research Hospital, Via Turi 27, 70013 Castellana Grotte, Bari, Italy; annamaria.cisternino@irccsdebellis.it; 5Laboratory of Nutritional Biochemistry (MN), National Gastroenterology Institute, “S. de Bellis” Research Hospital, Via Turi 27, 70013 Castellana Grotte, Bari, Italy; maria.notarnicola@irccsdebellis.it

**Keywords:** high adherence, lifestyle score, rate of mortality, sub-distribution hazard ratio

## Abstract

Background: In real life, nutrition goes beyond purely biological domains. Primary prevention is the most efficient approach for reducing the risk of mortality. We aimed to study the association of lifestyle, as measured by a modified World Cancer Research Fund/American Institute for Cancer Research (mWCRF/AICR) scoring system with all-cause, digestive system disease-related (DSD-related), cardiovascular disease-related (CVD-related), cancer–related and other cause-related mortality using data from two population-based cohort studies conducted in Southern Italy. Methods: A random sample of 5271 subjects aged 18 years or older was enrolled in 2005–2006 and followed up until 2020. Usual food intakes were estimated using a validated dietary questionnaire. Competing risks survival models were applied. Results: High adherence to the mWCRF/AICR score was found to be statistically significant and negatively associated with all-cause mortality (HR 0.56, 95%CI 0.39; 0.82), DSD-related mortality (SHR 0.38, 95%CI 0.15; 0.97) and cancer-related mortality (SHR 0.43, 95%CI 0.19; 0.97) in the male sub-cohort and other-cause mortality (SHR 0.43, 95%CI 0.21; 0.88) only in the female group. Conclusions: This mWCRF/AICR score can be seen as a simple, easy tool for use in clinical practice to evaluate both qualitative and quantitative aspects of the diet.

## 1. Introduction

A decreased morbidity and improvements in the desired quality of life can be achieved in a population by means of health promotion, when this takes deep root in the consciousness of that population [1]. Scientific efforts to elucidate the relationship between nutrition and health have greatly improved our understanding of the association between diet and health. Nutritional conditions in real life, in healthy individuals who have an adequate diet, do not depend only on individual ingredients or products, but also on a correct understanding of the idea of a “balanced diet”, since the human metabolism features a great capacity for flexibility [2]. The relevance of nutrition science lies primarily in the growing knowledge of the long-term impact of nutrients, foods and eating patterns on both health maintenance and disease onset [3]. This requires studies to be expanded to adjacent scientific fields beyond biomedical domains, such as social sciences and data sciences, in order to better understand what drives humans to desire the foods they eat. In fact, differential mortality rates have been described, associated with social inequality and consequently unlike lifestyles which include different ways of eating [4]. Nutrition sciences are not only about the biochemical aspects, but also include cultural and behavioral elements, as well as environmental sustainability issues [5].

Scientific challenges promote the development of Transdisciplinary Research/approaches and open science data (FAIR: Findable, Accessible, Interoperable and Reusable) [6].

Humans are currently facing a global transition in food production [7,8], and future breakthroughs in nutritional science will be strategic. The indications of the World Cancer Research Fund are a reference not only for a correct diet, but also for physical activity, in the prevention of oncological diseases [9]. These indications are also considered a valid prevention tool for chronic diseases with risk factors related to eating habits [10,11,12], and have been used in several observational studies in different populations [13,14,15,16]. The association of WCRF/AIRC recommendations with all-cause, and cause-specific mortality, such as cancer mortality, has been extensively studied in several geographical areas [11,17,18,19]. This association has been explored in a variety of sites including colon, breast and pancreatic cancers [19]. WCRF/AIRC recommendations and lower all-cause mortality rate is the most prevalent association documented in literature [17] whereas an association has not always been found with cancer cause-specific mortality [18]. The beneficial effects of adherence to one or more WCRF/AIRC components of the recommendations have also been observed between recent and long-term cancer survivors [19]. However, a comparative study of six dietary indexes conducted in Iran did not find any association between WCRF/AIRC recommendations and cancer mortality [11], nor did a recent study from an area similar to ours find an association between Mediterranean diet and cancer mortality [20].

It is interesting to note that most studies aimed at probing the association between WCRF/AIRC recommendations and mortality relied on Cox’s survival model for all the associations considered including cause-specific mortality [11,17,20]. Cause-specific mortality is a typical example of competing risks that frequently occur in epidemiologic studies but are often not recognized or ignored [21,22]. The use of classical survival analysis to estimate the incidence function and sub-distribution hazards ratios (SHR) may result in upward biased estimates [23]. Then, an appropriate statistical methodology should be applied.

Mediterranean diet is the most prevalent dietary pattern in this geographical area and its dietary components as well other recommendations included recently in the Mediterranean diet pyramid [24], closely fit WCRF/AIRC recommendations [25]. Furthermore, the adherence to Mediterranean diet in this area seems to have changed little over time, but with a differential adherence between sexes [26].

Our Institution, the National Institute of Gastroenterology ‘S de Bellis’ Research Hospital has conducted several epidemiological studies in this area and has documented a negative high age-related prevalence and a low incidence of Hepatitis C Virus infection. This infection produces a wide spectrum of gastrointestinal diseases from simple hepatitis to hepatocellular carcinoma. We decided that considering digestive disease-related (DSD) deaths rather than only digestive system cancers could more faithfully represent the cause-specific mortality in our study. This approach is further reinforced by the fact that only 19.4% and 7.2% of deaths were cancers and digestive system cancers, respectively [27]. For this purpose, we built up a modified WCRF/AICR score (mWCRF/AICR), following the WCRF/AICR indications, introducing some changes related to our study population [9,16,28,29,30].

This prospective cohort study conducted in Southern Italy during the period 2005–2020 was aimed at estimating the association of adherence to a mWCRF/AICR score with DSD-related, cardiovascular disease (CVD)-related, cancer-related and other-cause-related mortality. The score is intended as a tool for investigating a balanced and healthy diet, from the perspective described above, which can be readily used as an investigation tool also in clinical practice, in order to promote dietary and lifestyle behaviors aimed at maintaining a state of good health.

## 2. Materials and Methods

Details about the study population have been published elsewhere [31,32]. Briefly, two different prospective cohort studies conducted by the Laboratory of Epidemiology and Biostatistics of the National Institute of Gastroenterology, “Saverio de Bellis” Research Hospital (Castellana Grotte, Bari, Italy) were included. The Multicentrica Italiana Colelitiasi (MICOL) Study [33], is a population-based prospective cohort study of subjects, randomly drawn from the electoral list of Castellana Grotte (≥30 years old) in 1985 and followed up in 1992, 2005–2006 and 2013–2016. In 2005–2006, this cohort was added with a random sample of subjects from the PANEL study, aged 30–50 years, to compensate for the cohort aging. In this paper, the baseline for the MICOL cohort was established in 2005–2006 to capture all ages and to homogenize follow-up time.

The Nutrition Hepatology (NUTRIHEP) Study is a cohort of subjects enrolled in 2005–2006 from the city of Putignano (Apulia, Southern Italy). Using a systematic random 1-in-5 sampling procedure, a sample of the general population > 18 years old was drawn from the General Practitioner’s list of records. We used the records of General Practitioners, instead of drawing from the census, because no significant difference was found between the distribution of the general population from Putignano and the subjects inscribed in GPs’ records. In Italy, it is stated by law that everybody should have a GP. Therefore, the general population lists in the GP offices corresponds to the census list. A possible selection bias lies in the fact that specific sub-cohorts of patients (i.e., senior patients, patients with chronic diseases, patients with known chronic liver disease) would be more likely to be seen by the GP over a limited period of time. To minimize this potential confounding, a statistical analysis was carried out to test whether the mean age of the general population was comparable with that of subjects recruited by GP clinics. Therefore, we used one-way analysis of variance (ANOVA) and Bonferroni’s test for multiple comparisons. ANOVA was then used to test the hypothesis that sex-specific mean age was the same among the general population and subjects of the GP clinics. There was no statistical evidence on differences in the mean age (*p* = 0.15).

Considering the 2005–2006 period as the baseline for both studies, a total of 6114 subjects was invited to participate.

For the Micol/Panel cohort, a total of 3614 subjects was invited to participate (of which 1708 were from the Micol study and 1906 from the Panel study). Of these, 2970 (82.2% response rate) agreed to participate. We excluded 122 subjects for incomplete information (110 missing the food frequency questionnaire and 12 missing Body Mass Index measurements). Finally, 2848 subjects (78.8% inclusion rate) were included in the study.

For the NUTRIHEP cohort, 2500 persons were invited to participate and 2301 agreed (92% response rate). We excluded 283 subjects for incomplete information (256 missing the food frequency questionnaire and 27 missing Body Mass Index measurements), thus 2018 subjects (80.7% inclusion rate) were actually included in the study

Therefore, 5271 out of 6114 (86.2% response rate) agreed to participate, and, since for 405 subjects we did not have complete information, 4866 out of 6114 (79.6% inclusion rate) subjects were finally included. All procedures were performed in accordance with the ethical standards of the institutional research committee (IRCCS Saverio de Bellis Research) and Ethical Committee approval for the MICOL Study (DDG-CE-347/1984; DDG-CE-453/1991; DDG-CE-589/2004; DDG-CE 782/2013); and the NUTRIHEP Study in 2005 and 2014 (DDG-CE-502/2005; DDG-CE-792/2014), and with the 1964 Helsinki declaration and its later amendments.

### 2.1. Data Collection

Participants were interviewed to collect information on sociodemographic characteristics, health status, personal history and lifestyle factors including history of tobacco use (never, former—quit 5 or more years before—and current), food intake, education (illiterate, primary school, secondary school, high school, graduate) [34], job (managers and professionals, craft, agricultural and sales workers, elementary occupations, housewives, pensioners and jobless) [35] and marital status (single, married/coupled, separated/divorced and widow/er).

Weight was taken with the subject in underwear, standing on an electronic balance, SECA^®^, and was approximated to the nearest 0.1 kg. Height was measured with a wall-mounted stadiometer SECA^®^, approximated to 1 cm. Blood pressure (BP) measurement was performed following international guidelines [36]. The average of 3 BP measurements was calculated.

The European Prospective Investigation on Cancer (EPIC) Food Frequency Questionnaire (FFQ) [37], was administered by trained nutritionists to estimate the usual food intakes. Individual nutrient intakes were derived from foods included in the dietary questionnaires through the standardized EPIC Nutrient Database [38,39]. The EPIC FFQ input was performed online, and centralized processing was carried out by the National Cancer Institute, based in Milan.

Fasting venous blood samples were drawn, and the serum was separated into two different aliquots. One aliquot was immediately stored at −80 °C. The second aliquot was used to test biochemical serum markers by standard laboratory techniques in our Central laboratory.

### 2.2. Exposure Assessment

Adherence to the WCRF/AICR indications was estimated with the WCRF/AICR score. The scoring system was built up referring to the WCRF/AICR indications applied to EpiGEICAM data [14]. The score consisted of 11 items linked to the following domains: (a) maintain adequate body weight, (b) limit the consumption of energy-dense foods, (c) eat mostly foods of plant origin, (d) limit the intake of red meat and avoid processed meat, (e) limit alcoholic drinks and the consumption of salt and salt-preserved foods. For each of 11 items a maximum score of 1 was assigned when the recommendations were fully satisfied, a value of 0 when the recommendations were not satisfied and 0.5 points as an intermediate score. Higher scores indicate a greater concordance with the WCRF/AICR recommendations [13,14].

We were not able to evaluate weight history, physical activity and dietary supplement use because this information was not available.

### 2.3. Tracing Procedures and Outcome Assessment

Information about vital status of participants was obtained from the Municipalities of Castellana Grotte and Putignano, and electronically linked with the database. Inquiries were also made at the Municipalities of current residence of subjects who had moved. Tracing procedures and outcome assessment information on the underlying cause of death was extracted from the Apulian Regional Registry using the death certificate based on WHO guidelines [40]. The subject’s municipality of residence was queried if there were inconsistencies, then the cause of death was coded. The International Classification of Diseases, 10th edition (ICD-10) was used. Deaths were coded as follows: digestive system disease-related mortality (DSD-related mortality) (ICD-10 B17–18, C15–26, K55–92), cardiovascular disease-related mortality (CVD-related mortality) (ICD-10 I00-I99), cancer-related mortality (ICD C00–C14, C27–C97) and other-causes mortality (remaining ICD-10 codes). It was possible to trace the vital status for all included persons.

### 2.4. Statistical Analysis

Cut-off points to assign the score to each of 11 items assessed were taken from the literature [14]. For analytical purposes, the index was grouped in 3 categories: ≤5 (Low Adherence), 5–7 (Medium Adherence) and >7 (High Adherence) points

Data are presented as mean (±SD), median (±IQR) for continuous data or frequency (%) for categorical data. ANOVA and Pearson’s chi-squared tests were used to test differences between means and proportions, respectively.

Time from enrollment to death, moving elsewhere or end of the study (31 December 2020), whichever occurred first, was the observation time. The logarithm of the hazard of the outcome was modelled as a function of the baseline hazard and selected explanatory variables which may vary with time. and the following covariates, namely: sex, smoking never/former (quit smoking 5 years before or more) vs. current, hypertension, triglycerides (in range vs. non in range), glucose, glutamic-pyruvate transaminase (GPT), γ-Glutamyltransferase (GGT), body mass index (BMI), marital status (single, married/coupled, separated/divorced, widower) and education and job. Missing values were handled as an independent category in the variable.

A Cox’s model was fitted to the data to assess the association between mWCRF/AICR and all-cause mortality [41]. Proportional hazard assumption was performed by visual inspection of the -ln{-ln(survival)} curves and tested by using Schoenfeld residuals.

For cause-specific mortality, proportional hazards models for subdistributions were performed using a competing risks approach [42]. We estimated the Sub-Distribution Hazard Ratio (SHR), which reflects the association of the mWCRF/AICR score with the risk of developing four types of competing events: DSD-related mortality, CVD-related mortality, cancer-related mortality and other-cause mortality. Finally, by using post-estimation tools we predicted the SDHs by cause and sex [43]. In the modelling step, a likelihood ratio test was performed to include or not include each variable.

Since age is the most important risk factor for death, we chose age at death as the time scale.

Before fitting survival models, to overcome the problem of a few events in the tails of age at death distribution, we performed a separate analysis to establish the best range of age at death. In this analysis, we applied Cox’s Model and a Proportional Hazards Model for the subdistribution of competing events to the data, using different ranges of age at death and mWCRF/AICR score for the whole cohort and sex. Using Akaike’s (AIC) and Schwarz’s Bayesian information (BIC) criteria, we chose the best performing model. In all analyses we considered the comparison between High Adherence of mWCRF/AICR score vs. Low Adherence (>7 vs. ≤5.)

All statistical analyses were performed using Stata, statistical software version 16.1 (StataCorp, 4905 Lakeway Drive, College Station, TX 77,845, USA); in particular, the command and post-estimation commands from the stcrreg official Stata command were used. A *p* value ≤ 0.05 was considered as statistically significant.

## 3. Results

### 3.1. Modified WCRF/AICR Score Components

Percentages of participants who scored the maximum for the mWCRF/AICR score by sex and age are shown in Table 1. About 70% of participants scored the maximum for fruit and vegetable consumption, whereas about 75% of participants scored the maximum for sodium level. In particular, among females, 79.2% had a sodium intake of less than 2.4 g/day; 80% consumed no more than 10 g/day of alcohol; 35.2% consumed no more than 7 g/day of cold meats; 29.2% consumed less than 500 g of red meat and less than 3 g/day of processed meat per week and 59.9% had an intake of less than 91 g/day of white bread, pasta and rice. Seventy-one percent of males had a sodium intake of less than 2.4 g/day; about 45% consumed no more than 10 g/day of alcohol; 22.2% consumed less than 500 g of red meat and less than 3 g/day of processed meat per week; 27.7% consumed no more than 7 g/day and 39.6 had an intake of less than 91g/day of refined grains and refined grain products.

### 3.2. The Cohort

Appendix A show the flowchart of the study. There was complete information about 4866 (79.6%) participants. The study base generated a total observation time of 68,817.42 person years. Baseline characteristics of participants are shown in Table 2.

Higher scorers were more likely to be older and women; they were also more likely to be married, with secondary or higher educational level and pensioners. Descriptions of each cohort, as well as a comparison among subjects with complete and incomplete data, are shown in Appendix A. Missing data are referred exclusively to subjects who did not return the FFQ, and for which BMI measurements were missing.

Sensitivity analysis to obtain the best age range at death is shown for all-cause mortality and for cause-specific mortality, respectively, in Appendix A: the most suitable age range at death with a good trade-off between estimates and AIC/BIC criterion was 30–90. Number of deaths and mortality rates with their corresponding 95%CI for the whole cohort and by sex for all-cause and cause-specific mortality are shown in Appendix A. Results from the Proportional Hazards Model for Subdistribution of a Competing Risk are shown in Table 3.

A negative trend from High to Low Adherence to mWRCF/AICR score was observed (HR 0.92, 95%CI 0.86; 0.97) in the whole sample for all-cause mortality. A similar negative trend was observed in the male sub-cohort (HR 0.91, 95%CI 0.84; 0.98) and also a negative effect of High Adherence to mWRCF/AICR score (HR 0.56, 95%CI 0.39; 0.82).

A negative effect of High Adherence to mWCRF/AICR score was observed in the male sub-cohort for DSD_related mortality (SHR 0.38, 95%CI 0.15; 0.97) and a negative trend (HR 0.83, 95%CI 0.70; 0.99) for cancer-related mortality; moreover, there were negative effects of Medium Adherence (SHR 0.54, 95%CI 0.30; 0.85) and High Adherence (SHR 0.43, 95%CI 0.19; 0.97) as well as a negative trend (HR 0.83 95%CI 0.70; 0.99). For other-cause mortality, there were negative trends in the whole sample (HR 0.90, 95% CI 0.82; 0.99) and the female sub-cohort (HR 0.85, 95%CI 0.73; 0.99). There was also a negative effect of High Adherence in the female sub-cohort (SHR 0.43, 95%CI 0.21; 0.88).

Results about the modelling process are graphically represented in Figure 1. In the male sub-cohort for both DSD-related mortality and Cancer-related mortality, it is noted that the two SHR curves relate to adherence to the mWCRF/AICR score tend to maintain an equidistance between them over the time and the SHR curve related to High Adeherence is always lower than the other one. In particular, for cancer-related mortality, the SHR seems to decrease over time.

In the female sub-cohort for other-causes mortality, it is noteworthy that the three curves referred to adherence to the mWCRF/AICR score tend to maintain a certain equidistance, which is constant over time, and curves related to maximum adherence are always below the other two.

## 4. Discussion

In this cohort study conducted in southern Italy where the Mediterranean Diet is most prevalent way of eating, high adherence to mWCRF/AICR score showed an important protective effect on all-cause mortality in the male sub-cohort as well as, in particular, cause-specific mortality scenarios. DSD-related mortality decreased when the adherence to mWCRF/AICR score was the highest in the male sub-cohort, as well as for cancer-related mortality. Instead, for other-cause mortality, the highest adherence to the mWCRF/AICR score resulted in a significant reduction in the risk of mortality in the females’ sub-cohort. No statistically significant association emerged between adherence to the mWCRF/AICR score and CVD-related mortality

The association between a healthy diet, with a reduced risk of all-causes mortality, and the incidence of major chronic diseases has been shown in large epidemiologic studies [44,45]. Estimated effects of the Mediterranean diet, such as reduction in all-causes mortality and cause-specific mortality [46,47], reduced incidence of cardiovascular and cerebrovascular diseases, reduced incidence of neoplastic diseases and neurodegenerative diseases as well as other clinical outcomes, such as stroke and mild cognitive disorders, which have been reported [48]. Furthermore, results of the SU.VI.MAX trial suggest that antioxidants may contribute to counteract some of the potential deleterious effects of a pro-inflammatory diet on mortality [49]. The Mediterranean style with its low inflammatory potential may be associated with better outcomes [50]. However, we did not find any association between adherence to mWCRF/AICR score and CVD-related mortality. A meta-analysis about MedDiet individual components and CVD-related mortality evidenced a protective role of olive oil, fruits, vegetables and legumes [51] This contrasting result may reflect the intensive intake of the MedDiet components aforementioned in this geographical area, resulting in a homogeneous distribution of the exposure. Non-communicable diseases (NCD) are the main cause of death in developed countries [52]. Changes in lifestyles, especially diet, could play an essential role in preventing NCD and premature mortality [53]. However, social inequalities may play an important role as determinant of mortality. It has been observed an inverse relationship between attained educational level and mortality rates [4]. This could imply different adherence to more healthy lifestyle. Dietary patterns associated with a lack of compliance with WCRF/AICR recommendations, have been associated with higher concentration of inflammatory markers [54], and a higher intake of fruit and vegetables is associated with a lower mortality. Indeed, in a recent study the risk reduction showed a plateau at ≈5 servings of fruit and vegetables per day. These findings support current dietary recommendations to increase the intake of fruits and vegetables [55]. In the EPIC study, participants reporting the consumption of more than 569 g/day of fruits and vegetables, had lower risks of death from diseases of the circulatory, respiratory and digestive system, when compared with participants consuming less than 249 g/day. The lower risk of death associated with a higher consumption of fruits and vegetables may be the result of inverse associations with diseases of the circulatory, respiratory and digestive systems [56]. Emerging evidence has shown that lifestyle, including diet, after a Colorectal Cancer (CRC) diagnosis might affect all-cause and CRC-specific mortality risk, in particular, in terms of risk of relapse, mortality [57] and survival [58]. Other studies have highlighted that lifestyle and eating habits are associated with gastric adenocarcinoma [59].

Our results show that most of the participants with the highest total score had a high daily intake of fruit and vegetables. The findings are consistent with the consensus that plant-based diets are beneficial for health, as Burkitt had already hypothesized in 1969 [60]. Low fiber consumption in high-income countries could be linked to the high prevalence of western diseases in those populations [61]. Current recommendations for dietary fiber intake for adults in most European countries and for countries such as Australia, New Zealand and the USA are between 30–35 g per day for males and between 25–32 g per day for females [62]. These findings were already confirmed in the EPIC cohort [63]. Fiber intake is correlated with the occurrence of cancer and diabetes, but also with all-causes and cause-specific cardiovascular mortality. Dietary fiber is known to (1) improve laxation by increasing bulk and reducing the transit time of feces through the bowel; (2) increase the excretion of bile acids, estrogen and fecal pro-carcinogens and carcinogens by binding to them; (3) lower serum cholesterol; (4) slow glucose absorption and improve insulin sensitivity; (5) lower blood pressure; (6) promote weight loss; (7) inhibit lipid peroxidation; and (8) have anti-inflammatory properties [64]. Moreover, mechanistic studies have shown that products of fiber fermentation in the colon could suppress colonic mucosal inflammation and carcinogenesis [65]. Short-chain fatty acids can affect the epigenome through metabolic regulatory receptors, and this can reduce obesity, diabetes, atherosclerosis, allergy and cancer [66]. Moreover, studies conducted in different countries, highlighted strong associations between the consumption of ultra-processed foods and an increased risk of obesity and several other diet-related chronic diseases [67]. Increasing dietary fiber intake to 50 g/day is likely to increase the lifespan, improve the quality of life during the added years and substantially reduce healthcare costs. The recommendation to eat mostly foods of plant origin is included among the WCFR/AICR recommendations [9].

A more general adoption of plant-based diets could lead to benefits also for planetary health [68]. The Mediterranean diet is considered to be one of the environmentally friendly options [69]. The amount of animal-based foods in the diet, particularly meat and dairy products, is the most significant contributor to the harmful effects for the environment and to a suboptimal sustainability [70]. As the diet influences not only health but also the environment, dietary advice should take into account the environmental impact of the diet. An enhanced adherence to an eco-friendly diet like the Mediterranean diet is an important goal in society today [69]. New environmental dimensions have been included in the Mediterranean Diet Pyramid. They highlight food intake recommendations and address both health and environmental issues emphasizing a lower consumption of red meat and bovine dairy products, and a higher consumption of legumes and locally grown eco-friendly plant foods [24].

Our study showed that for DSD-related mortality and for cancer-related mortality, only in the male sub-cohort did an association emerge between a high score and a reduction in mortality risk. As well as for other-cause mortality in the female sub-cohort, gender differences have been reported also for dietary habits, as well as individual responses to dietary intake [71,72]. Indeed, there appears to be some resistance to following a healthy diet in males [73]. In fact, in our region adherence to a healthier lifestyle including dietary habits is higher among females and this adherence has changed little over time [26]. Furthermore, the decreased risk of dying of cancer beyond 60 years or older among males may reflect the exhaustion of the causal model of cancer [74]. Masculinities is a term now commonly used to denote diversity and complexity among males and forms of masculine identity; there are suggestions that conventional masculinities play a negative role in male health [75]. In a Scandinavian study which specifically considered males, masculinities and food preferences, several professional groups were sampled, and it was found that the working-class males, carpenters, interpreted food in terms of fuel and rejected the traditional associations between food and health. On the contrary, the middle-class participants, professional engineers, saw food more in terms of pleasure and of enjoying good food and a good life [76]. Moreover, a Polish study has evidenced a differential decrease in mortality rates being more intense in males. Additionally, a decrease in cardiovascular and lung cancer mortality rates in males and an increase in mortality rates from suicides and lung cancer in females were observed [77].

Our study has several strengths, particularly the cohort design and the large random population sample from a geographic area where the Med Diet is widespread. Moreover, the complete exposure assessment was performed using a recognized score [14]. In a 1997 validity study conducted by the EPIC study group in Italy, they compared results of the FFQ with 24 h recall diet assessment. Subsequent modifications were made to the questionnaire by the EPIC study team, expected to decrease measurement errors. Our study did not include a measure of physical activity, a potentially serious limitation given previous research findings linking physical activity with all-causes and cause-specific mortality. Thus, we may have over- or underestimated the effect of diet due to a confounding or effect modification of physical activity [78]. Our score refers to the 2007 WCRF/AICR recommendations for the prevention of cancer; compared with those issued in 2017, the recommendation to limit the consumption of salt was more specific in the latter.

## 5. Conclusions

Primary prevention is the most effective and economical approach to prevent chronic diseases. Emphasizing the quality and quantity of the diet may be the best preventive measure to achieve long-term personal and social goals at every stage of life. The healthy life span loss is evident from the myriad non-communicable diseases that are insidious and manifest with sudden cardiovascular events, liver failure, lung disease, diabetes and all types of tumors. Our study showed a reduction in the risk of mortality for digestive system disease, in male subjects who had a high adherence to the score as well as cancer-related mortality. Additionally, a reduction in mortality was observed for all other-causes mortality for females who had a high adherence to the score. Further in-depth studies would be useful to evaluate possible educational interventions to promote a healthy lifestyle in vulnerable groups of the population in the reference area

The novelty of this paper consists of our mWCRF/AICR score, applied to a Mediterranean population with an appropriate statistical methodology aimed at obtaining valid and precise estimates. It is an easy-to-use tool in clinical practice that allows a simple evaluation of both the qualitative and quantitative aspects of the diet, as well as complete lifestyle. It could also be used by patients themselves as an immediate self-assessment tool, aimed at fostering a greater awareness of their lifestyle habits. Last but not least, our score has a cross-border character and could be used to compare lifestyle habits in different populations.

## Figures and Tables

**Figure 1 nutrients-13-04002-f001:**
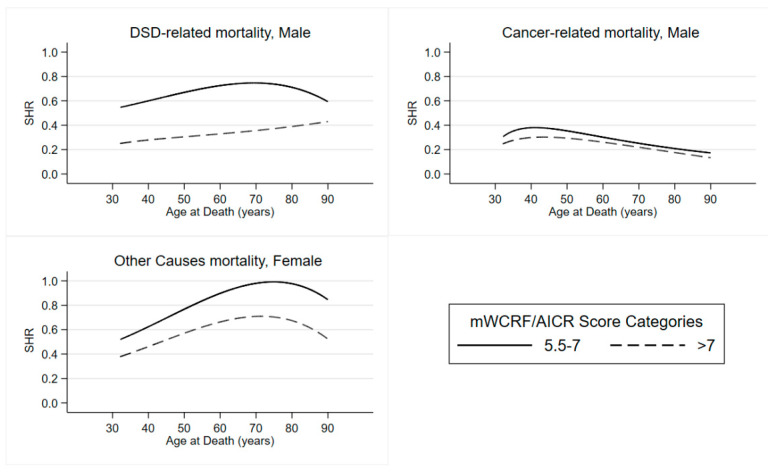
DSD-related mortality, Cancer-related mortality and Other Causes mortality Subdistribution Hazard Ratio. MICOL/PANEL and NUTRIHEP Studies. Castellana Grotte-Putignano (BA), Italy, 2005-2020. DSD: Digestive System Disease; SHR: Subdistribution Hazard Ratio.

**Table 1 nutrients-13-04002-t001:** Scoring criteria of the modified WCRF/AICR score components and percentage of participants with maximum score.

	Scoring Criteria of Modified WCRF/AICR Score			% of Participants with Maximum Component Scores							
Components				Age Class (Years)					Sex		
	0 ^#^	0.5 ^ǂ^	1 ^§^	<40 *n* = 1322	41–49 *n* = 1024	50–59 *n* = 996	60–69 *n* = 792	≥70 *n* = 732	Female *n* = 2352	Male *n* = 2514	Total *n* = 4866
Energy dense foods (kcal/100g)	>175	125–175	≤125	30.83	27.14	28.84	23.63	18.61	33.16	20.64	26.70
Fast food intake (g/day)	>42	18–42	<18	1.25	1.61	5.61	14.30	20.69	7.70	6.76	7.21
Sugary drinks intake (g/day)	>250	≤250	0	6.78	14.53	24.73	36.53	44.44	23.04	21.93	22.47
Fruits and Vegetables (g)	<200	200–400	≥400	63.20	71.44	71.16	73.18	74.72	69.77	69.93	69.85
Cereals. Whole grain bread and Legumes (g)	<24	24–64	≥64	27.21	19.88	20.57	12.64	10.83	20.66	18.54	19.56
White bread. pasta and rice (g/day)	≥144	91–144	<91	39.68	43.69	54.92	57.60	58.75	59.86	39.58	49.38
* Red (R) and processed meat (P)	R+P ≥ 500 or P ≥ 50	R+P < 500 and P 3–50	R+P < 500 and P < 3	13.50	18.26	24.41	37.55	47.22	29.21	22.24	25.61
** Cold meat (g/day)	>22	7–22	≤7	14.97	19.98	31.79	48.02	59.03	35.25	27.68	31.34
Alcohol intake (g/day)	≥20	10–20	≤10	73.01	62.97	54.23	53.64	60.56	80.23	45.11	62.08
Sodium (g/day)	≥3	2.4–3	<2.4	62.68	71.36	79.97	84.36	88.34	79.17	71.04	74.98
*** BMI (kg/m^2^)	<18.5 and ≥30	25–30	18.5–24.9	52.80	36.73	23.33	16.86	18.33	39.33	26.09	32.49

^§^ Maximum score 1 was assigned when the recommendations were fully satisfied; ^ǂ^ Intermediate score 0.5 when the recommendations were partially satisfied; ^#^ Low score 0 when the recommendations were partially satisfied. * Red and Processed meat g/week; Processed meat g/day; ** Cold meat: meats subjected to a salting process; *** BMI: Body Mass Index.

**Table 2 nutrients-13-04002-t002:** Characteristics of participants by modified WCRF/AICR score categories. MICOL/PANEL and NUTRIHEP Studies. Castellana Grotte. Putignano (BA). Italy. 2005–2020.

	All ^¥^	Modified WCRF/AICR Score Categories			
	0–11	≤5	5.5–7	>7	*p*-Value
*n* ***	4866	771	2949	1146	
Age at Enrollment (years) *	51.46 (15.81)	48.90 (14.19)	50.92 (15.73)	54.56 (16.57)	<0.001
Age (categorical. years) ***					<0.001
<40	1356 (27.9)	227 (29.4)	869 (29.5)	260 (22.7)	
40–49	991 (20.4)	199 (25.8)	592 (20.1)	200 (17.5)	
50–59	1016 (20.9)	176 (22.8)	608 (20.6)	232 (20.2)	
60–69	784 (16.1)	96 (12.5)	475 (16.1)	213 (18.6)	
≥70	719 (14.8)	73 (9.5)	405 (13.7)	241 (21.0)	
Sex ***					<0.001
Female	2352 (48.3)	147 (19.1)	1369 (46.4)	836 (72.9)	
Male	2514 (51.7)	624 (80.9)	1580 (53.6)	310 (27.1)	
Marital Status ***					<0.001
Single	764 (15.7)	114 (14.8)	479 (16.2)	171 (14.9)	
Married/Coupled	3673 (75.5)	609 (79.0)	2231 (75.7)	833 (72.7)	
Separated/Divorced	117 (2.4)	19 (2.5)	68 (2.3)	30 (2.6)	
Widower	312 (6.4)	29 (3.8)	171 (5.8)	112 (9.8)	
Education ***					0.044
Primary School	1332 (27.4)	186 (24.1)	807 (27.4)	339 (29.6)	
Secondary School	1487 (30.6)	264 (34.2)	893 (30.3)	330 (28.8)	
High School	1385 (28.5)	224 (29.1)	833 (28.2)	328 (28.6)	
Graduated	493 (10.1)	70 (9.1)	321 (10.9)	102 (8.9)	
Illiterate	169 (3.5)	27 (3.5)	95 (3.2)	47 (4.1)	
Job ***					<0.001
Managers and Professionals	287 (5.9)	63 (8.2)	176 (6.0)	48 (4.2)	
Craft, Agricultural and Sales Workers	1279 (26.3)	227 (29.4)	765 (25.9)	287 (25.0)	
Elementary Occupations	1038 (21.3)	199 (25.8)	671 (22.8)	168 (14.7)	
Housewife	634 (13.0)	55 (7.1)	361 (12.2)	218 (19.0)	
Pensioneers	1372 (28.2)	196 (25.4)	815 (27.6)	361 (31.5)	
Jobless	254 (5.2)	31 (4.0)	160 (5.4)	63 (5.5)	
No Information	2 (<1)	0 (0.0)	1 (<1)	1 (0.1)	
DBP (mmHg) *	124.10 (17.81)	123.58 (16.70)	123.69 (17.98)	125.52 (18.06)	0.010
SBP (mmHg) *	76.76 (9.74)	77.50 (10.15)	76.38 (9.74)	77.24 (9.41)	0.003
Weight (kg) *	73.06 (14.97)	82.12 (15.18)	73.57 (14.63)	65.63 (11.62)	<0.001
BMI (kg/m^2^) *	27.51 (5.15)	29.55 (5.16)	27.65 (5.23)	25.77 (4.29)	<0.001
Kcal days	2182.34 (825.13)	2843.35 (998.83)	2200.11 (720.08)	1689.64 (588.49)	<0.001
Triglycerides (mmol/L) *	1.38 (0.98)	1.54 (1.13)	1.39 (0.99)	1.23 (0.81)	<0.001
Total Cholesterol (mmol/L) *	5.10 (1.01)	5.18 (1.00)	5.08 (1.00)	5.09 (1.06)	0.064
HDL (mmol/L) *	1.33 (0.35)	1.26 (0.32)	1.33 (0.36)	1.41 (0.36)	<0.001
LDL (mmol/L) *	3.14 (0.87)	3.22 (0.84)	3.13 (0.86)	3.13 (0.92)	0.038
Glucose (mmol/L) *	5.87 (1.40)	6.02 (1.26)	5.85 (1.32)	5.83 (1.65)	0.005
GPT (μkat/L) *	0.28 (0.22)	0.32 (0.22)	0.28 (0.21)	0.26 (0.25)	<0.001
GGT (μkat/L) *	0.25 (0.25)	0.31 (0.30)	0.25 (0.24)	0.22 (0.24)	<0.001
Smoke ***					<0.001
Never/Former	4029 (82.8)	603 (78.2)	2425 (82.2)	1001 (87.3)	
Current	837 (17.2)	168 (21.8)	524 (17.8)	145 (12.7)	
Observation time **	14.86 (14.20. 15.10)	14.89 (14.50. 15.20)	14.86 (14.20. 15.14)	14.80 (14.17. 14.97)	<0.001
Age at Death (years) **	65.74 (53.51. 77.17)	62.38 (53.31. 72.97)	64.47 (52.91. 76.68)	70.06 (55.90. 81.11)	<0.001
Status ***					0.008
Alive and/or Censored	4132 (84.9)	675 (87.5)	2512 (85.2)	945 (82.5)	
Dead	734 (15.1)	96 (12.5)	437 (14.8)	201 (17.5)	
Cause of Death ***					0.11
Alive and/or Censored	4132 (84.9)	675 (87.5)	2512 (85.2)	945 (82.5)	
DSD-related mortality	131 (2.7)	20 (2.6)	76 (2.6)	35 (3.1)	
CVD-related mortality	210 (4.3)	25 (3.2)	126 (4.3)	59 (5.1)	
CR-related mortality	128 (2.6)	21 (2.7)	77 (2.6)	30 (2.6)	
Other-Cause mortality	265 (5.4)	30 (3.9)	158 (5.4)	77 (6.7)	
Diabetes ***					0.43
No	4542 (93.3)	718 (93.1)	2763 (93.7)	1061 (92.6)	
Yes	324 (6.7)	53 (6.9)	186 (6.3)	85 (7.4)	
Dyslipidemia ***					0.011
No	4059 (83.4)	619 (80.3)	2460 (83.4)	980 (85.5)	
Yes	807 (16.6)	152 (19.7)	489 (16.6)	166 (14.5)	
Hypertension ***					0.91
No	3662 (75.3)	581 (75.4)	2224 (75.4)	857 (74.8)	
Yes	1204 (24.7)	190 (24.6)	725 (24.6)	289 (25.2)	

GPT: Glutamate Pyruvate Transaminase Alanine Aminotrasferase; GGT: γ-Glutamyltransferase; BMI: Body Mass Index; DBP: Diastolic Blood Pressure; SBP: Systolic Blood Pressure; HDL: High Density Lipoprotein Cholesterol; LDL: Low Density Lipoprotein Cholesterol; DSD-related mortality: digestive system disease-related mortality; CVD-related mortality: cardiovascular disease-related mortality; CR-related mortality: cancer-related mortality. Cells reporting subject characteristics contain * Mean ± (SD). ** Median (IQR). *** Number. ^¥^ (Percentage) Percentages calculated per column.

**Table 3 nutrients-13-04002-t003:** Hazard Ratio (HR) and 95% Confidence Intervals (95% CI) for all-cause mortality and Subdistribution Hazard Ratio (SHR) and 95% Confidence Intervals (95% CI) for DSD-related mortality, CVD-related mortality, CR-related mortality and other-cause mortality from modified WCRF/AICR.

	mWCRF/AICR Score Categories				mWCRF/AICR (Continuos)	
		5.5–7	>7			
	HR	95% CI	HR	95% CI	HR	95% CI
All-Cause mortality						
Whole Sample	0.94	0.74; 1.19	0.75	0.57; 1.00	0.92 *	0.86; 0.97
Female	1.39	0.77; 2.51	1.38	0.75; 2.52	0.95	0.86; 1.05
Male	0.86	0.65; 1.13	0.56 *	0.39; 0.82	0.91 *	0.84; 0.98
	SHR	95% CI	SHR	95% CI	SHR	95% CI
DSD-related mortality						
Whole Sample	0.82	0.50; 1.36	0.72	0.38; 1.37	0.98	0.86; 1.13
Female	1.59	0.35; 7.09	2.62	0.62; 11.00	1.20	0.92; 1.58
Male	0.79	0.46; 1.36	0.38 *	0.15; 0.97	0.94	0.80; 1.10
CVD-related mortality						
Whole Sample	1.17	0.72; 1.90	1.19	0.69; 2.07	0.99	0.88; 1.12
Female	3.66	0.55; 24.3	4.14	0.61; 28.29	1.07	0.93; 1.24
Male	1.10	0.65; 1.86	1.01	0.51; 1.98	0.90	0.75; 1.08
Cancer-related mortality						
Whole Sample	0.76	0.46; 1.25	0.64	0.36; 1.14	0.89	0.78; 1.01
Female	2.36	0.52; 10.75	1.91	0.42; 8.68	0.93	0.76; 1.13
Male	0.54 *	0.30; 0.95	0.43 *	0.19; 0.97	0.83 *	0.70; 0.99
Other-Cause mortality						
Whole Sample	1.11	0.72; 1.72	0.87	0.52; 1.44	0.90 *	0.82; 0.99
Female	0.54	0.27; 1.08	0.43 *	0.21; 0.88	0.85 *	0.73; 0.99
Male	1.39	0.81; 2.39	1.12	0.58; 2.15	0.95	0.84; 1.08

<5 Referent category. Adjusted for: Sex, Smoking (Never/Former vs. Current), Hypertension, Triglycerides in range vs. non in range, Glucose (mmol/L), Glutamate Pyruvate Transaminase (μkat/L), γ-Glutamyltransferase (μkat/L), BMI: Body Mass Index (kg/m^2^), Marital Status; Education, Job. DSD-related mortality: digestive system disease-related mortality, CVD-related mortality: cardiovascular disease-related mortality, CR-related mortality: cancer-related mortality. * *p* < 0.05.

## Data Availability

The data described in the manuscript, in the code book and in the analytical code will be made available at the request of the reviewers.

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
