# Peer review of "Modified WCRF/AICR Score and All-Cause, Digestive System, Cardiovascular, Cancer and Other-Cause-Related Mortality: A Competing Risk Analysis of Two Cohort Studies Conducted in Southern Italy"

_nutrients, 2021, doi:10.3390/nu13114002_

Round 1

Reviewer 1 Report

Dear Authors,
Thank you very much for the opportunity to review this interesting paper. The manuscript describes a highly significant research problem, namely the relationship between the quality of lifestyle (diet) and life expectancy. The study was conducted on a large sample in a follow-up system with the use of adequate statistical methods. The supplemented and expanded version of the work is legible and satisfactory for the reader. I have a few minor comments only:

L. 305+Rates of premature mortality among adults are important measures of the economic and psychosocial well-being of human populations. In many countries, such rates are, as a rule, inversely related to the level of attained education. A lot of research points to a regular educational gradient in mortality (Social inequality in premature mortality among Polish urban adults during economic transition. Am J Hum Biol. 2007;19(6):878-85.) In this study, the researchers also confirm the significant importance of the level of education in terms of lifestyle and, perhaps, life expectancy. I suggest that this relationship is worth mentioning more clearly in the introduction/discussion.

L. 367+ Sexual differences in mortality (different mortality rates) should be discussed with the results of other studies. Part of the research describe the pattern of change in excess male mortality. In some 
countries, women (due to their lifestyle) are more likely to die from causes considered to be male (eg suicide, lung cancer), while male mortality from CVD decreases significantly (Decrease in sex difference 
in premature mortality during system transformation in Poland. J Biosoc Sci. 2008;40(2):297-312.) A thread to be completed in the discussion.

L. 34+ keywords should be different than in the title
L. 231+Table 1 needs to be re-edited to make it more readable.

Thank you for the opportunity to review this article.

Author Response

We would thank the reviewer for his/her valuable comments.

Reviewer 1

Dear Authors,
Thank you very much for the opportunity to review this interesting paper. The manuscript describes a highly significant research problem, namely the relationship between the quality of lifestyle (diet) and life expectancy. The study was conducted on a large sample in a follow-up system with the use of adequate statistical methods. The supplemented and expanded version of the work is legible and satisfactory for the reader. I have a few minor comments only:

  • 305+Rates of premature mortality among adults are important measures of the economic and psychosocial well-being of human populations. In many countries, such rates are, as a rule, inversely related to the level of attained education. A lot of research points to a regular educational gradient in mortality (Social inequality in premature mortality among Polish urban adults during economic transition. Am J Hum Biol. 2007;19(6):878-85.) In this study, the researchers also confirm the significant importance of the level of education in terms of lifestyle and, perhaps, life expectancy. I suggest that this relationship is worth mentioning more clearly in the introduction/discussion.

We have added in the introduction and discussion section some sentences. Now, they read:

In fact, differential mortality rates have been described associated with social inequality and consequently unlike lifestyles which include different ways of eating.

However, social inequalities may play an important role as determinant of mortality. It has been observed an inverse relationship between attained educational level and mortality rates. This could imply different adherence to more healthy lifestyle.

  • 367+ Sexual differences in mortality (different mortality rates) should be discussed with the results of other studies. Part of the research describe the pattern of change in excess male mortality. In some 
    countries, women (due to their lifestyle) are more likely to die from causes considered to be male (eg suicide, lung cancer), while male mortality from CVD decreases significantly (Decrease in sex difference 
    in premature mortality during system transformation in Poland. J Biosoc Sci. 2008;40(2):297-312.) A thread to be completed in the discussion.

We have added a paragraph in the discussion section. Now, it reads:

Moreover, a Polish study has evidenced a differential decrease in mortality rates being more intense in males. Additionally, a decrease of cardiovascular and lung cancer mortality rates in males and an increase in mortality rates from suicides and lung cancer in females were observed

  • 34+ keywords should be different than in the title

We have modified the key words. Now, they read:

Keywords: High adherence;  lifestyle score; rate of mortality;; Sub-disribution Hazard Ratio

  • 231+Table 1 needs to be re-edited to make it more readable.

We have added some description in table 1 to make it more readable. Now, it reads:

TABLE 1. Scoring criteria of the modified WCRF/AICR score components and percentage of participants with maximum score

Scoring Criteria of modified WCRF/AICR score

% of participants with maximum component scores

Components

Age Class (years)

Sex

0#

0.5Ç‚

<40

n=1322

41-49

n=1024

50-59

n=996

60-69

n=792

≥70

n=732

Female

n=2352

Male

n=2514

Total

n=4866

Energy dense foods (kcal/100g)

>175

125-175

≤125

30.83

27.14

28.84

23.63

18.61

33.16

20.64

26.70

Fast food intake (g/day)

>42

18-42

<18

1.25

1.61

5.61

14.30

20.69

7.70

6.76

7.21

Sugary drinks intake (g/day)

>250

≤250

0

6.78

14.53

24.73

36.53

44.44

23.04

21.93

22.47

Fruits and Vegetables (g)

<200

200-400

≥400

63.20

71.44

71.16

73.18

74.72

69.77

69.93

69.85

Cereals. Whole grain bread and Legumes

(g)

<24

24-64

≥64

27.21

19.88

20.57

12.64

10.83

20.66

18.54

19.56

White bread. pasta and rice (g/day)

≥144

91-144

<91

39.68

43.69

54.92

57.60

58.75

59.86

39.58

49.38

*Red (R) and processed meat (P)

R+P≥500 or P≥50

R+P<500 and P 3-50

R+P<500 and P<3

13.50

18.26

24.41

37.55

47.22

29.21

22.24

25.61

**Cold meat (g/day)

>22

7-22

≤7

14.97

19.98

31.79

48.02

59.03

35.25

27.68

31.34

Alcohol intake (g/day)

≥20

10-20

≤10

73.01

62.97

54.23

53.64

60.56

80.23

45.11

62.08

Sodium (g/day)

≥3

2.4-3

<2.4

62.68

71.36

79.97

84.36

88.34

79.17

71.04

74.98

***BMI (kg/m2)

<18.5

and≥30

25-30

18.5-24.9

52.80

36.73

23.33

16.86

18.33

39.33

26.09

32.49

  • Maximum score 1 was assigned when the recommendations were fully satisfied; Ç‚Intermediate score 0.5 when the recommendations were partially satisfied; #Low score 0 when the recommendations were partially satisfied

Reviewer 2 Report

The methodological part of the article, together with statistical part, is very well conducted and the whole article with discussion is both interesting and appropriately designed. The methodology is described in detailed with justification of using particular approaches and performing exactly those analyses which adequately refer to the problem. The Authors found no statistically significant association between adherence to the mWCRF/AICR score and CVD related mortality. I suggest to discuss this result in the light of the result of the paper: PMID: 26528631, DOI: 10.1080/10408398.2015.1107021, where was found that a Mediterranean dietary pattern is associated with lower risks of CVD incidence and mortality, including CHD and MI. I recommend to accept the paper only after very minor revision. Details I present below:

  1. Page 2, line 51 “transdisciplinary disciplines” - maybe “Transdisciplinary Research/approaches” to not repeat the same thing
  2. Methods, line 26: “A random sample of 5271 subjects aged 18 years or older was enrolled in 2005-06 and followed-up 26 until 2017.” But in line 88 I found : “This prospective cohort study conducted in Southern Italy during the period 2005-2020” – the point of censoring data was december 2020, so I’m not sure when finished follow up period. Please, verify and just explain.
  3. Line 185: “The time-to-event, was modelled as a function of the outcome and thefollowing covariates…” Please reformulate this sentence. In fact the logarithm of the hazard of the outcome is modelled as a function of the baseline hazard and selected explanatory variables which may vary with time. Additionally, please delete comma and insert space between “thefollowing” in the sentence if you use the similar structure .
  4. Line 197 please reformulate “We estimated the Sub-Distribution Hazard Ratio (SHR), that is the association of the mWCRF/AICR score on the risk of developing four types of competing events” into eg. “We estimated the Sub-Distribution Hazard Ratio (SHR), which reflects the association of the mWCRF/AICR score with the risk of developing four types of competing events”
  5. Seventy-one per cent – shouldn’t be „percent”?
  6. I suggest to insert in table 1 the meaning of the particular WCRF/AICR score: 0, 0.5 and 1 analogically as in the exposure assessment part that “maximum score of 1 was assigned when the recommendations were fully satisfied, a value of 0 when the recommendations were not satisfied and 0.5 points as an intermediate score” Maybe some description could be added under the table?
  7. Lines 248-250 “High adherence scorers were ≥60 years old (about 40%) and women (73%). In contrast, Low Adherence scorers were younger (<60 years old, about 80%) and male (81%). High Adherence scorers were also married (72.7%), had Secondary School or higher Education (66.3%) and pensioneers (31.5%).“  I suggest to reformulate text as to indicate that some characteristics were more frequent or dominated in e.g. in the group of people with the highest score. In present form it is slightly misleading despite percentages in brackets.
  8. Titles of supplementary materials – shouldn’t be Sensitivity instead of Sensibility, I'm not sure whether in a statistic both these terms are used interchangeably.

Author Response

We would thank the reviewer for his/her valuable comments.

Reviewer 2

Comments and Suggestions for Authors

The methodological part of the article, together with statistical part, is very well conducted and the whole article with discussion is both interesting and appropriately designed. The methodology is described in detailed with justification of using particular approaches and performing exactly those analyses which adequately refer to the problem.

  • The Authors found no statistically significant association between adherence to the mWCRF/AICR score and CVD related mortality. I suggest to discuss this result in the light of the result of the paper: PMID: 26528631, DOI: 10.1080/10408398.2015.1107021, where was found that a Mediterranean dietary pattern is associated with lower risks of CVD incidence and mortality, including CHD and MI.

We have added a paragraph in the discussion section. Now, it reads:

However, we did not find any association between adherence to mWCRF/AICR score and CVD-related mortality. A meta-analysis about MedDiet individual components and CVD-related mortality evidenced a protective role of olive oil, fruits, vegetables and legumes This contrasting result may reflect the intensive intake of the MedDiet components aforementioned in this geographical area resulting in an homogeneous distribution of the exposure.

  • Page 2, line 51 “transdisciplinary disciplines” - maybe “Transdisciplinary Research/approaches” to not repeat the same thing

We have changed the words. Now they read: Scientific challenges promote the development of Transdisciplinary Research/approaches and open science data (FAIR: Findable, Accessible, Interoperable and Reusable).

  • Methods, line 26: “A random sample of 5271 subjects aged 18 years or older was enrolled in 2005-06 and followed-up 26 until 2017.” But in line 88 I found : “This prospective cohort study conducted in Southern Italy during the period 2005-2020” – the point of censoring data was december 2020, so I’m not sure when finished follow up period. Please, verify and just explain.

We have changed the mistake which now reads: Methods: A random sample of 5271 subjects aged 18 years or older was enrolled in 2005-06 and followed-up until 2020

  • Line 185: “The time-to-event, was modelled as a function of the outcome and thefollowing covariates…” Please reformulate this sentence. In fact the logarithm of the hazard of the outcome is modelled as a function of the baseline hazard and selected explanatory variables which may vary with time. Additionally, please delete comma and insert space between “thefollowing” in the sentence if you use the similar structure .

We have added the suggestion by the reviewer. Now, it reads:

The logarithm of the hazard of the outcome was modelled as a function of the baseline hazard and selected explanatory variables which may vary with time. and the following covariates, namely:sex , smoking never/former (quit smoking 5 years before or more)vs current, hypertension, , triglycerides (in range vs non in range), glucose, glutamic-pyruvate transaminase (GPT), γ-Glutamyltransferase (GGT), body mass index (BMI), marital status (single, married/coupled, separated/divorced,widower)and education and job. Missing values were handled as an independent category in the variable

  • Line 197 please reformulate “We estimated the Sub-Distribution Hazard Ratio (SHR), that is the association of the mWCRF/AICR score on the risk of developing four types of competing events” into eg. “We estimated the Sub-Distribution Hazard Ratio (SHR), which reflects the association of the mWCRF/AICR score with the risk of developing four types of competing events”

We have added the suggestion by the reviewer. Now, it reads:

We estimated the Sub-Distribution Hazard Ratio (SHR), which reflects the associationof the mWCRF/AICR score with the risk of developing four types of competing events: DSD-related mortality, CVD-related mortality, Cancer-related mortality and Other-Cause mortality.

  • Seventy-one per cent – shouldn’t be „percent”?

We have changed this typo

  • I suggest to insert in table 1 the meaning of the particular WCRF/AICR score: 0, 0.5 and 1 analogically as in the exposure assessment part that “maximum score of 1 was assigned when the recommendations were fully satisfied, a value of 0 when the recommendations were not satisfied and 0.5 points as an intermediate score” Maybe some description could be added under the table?

We have added some description in the footnote.

TABLE 1. Scoring criteria of the modified WCRF/AICR score components and percentage of participants with maximum score

Scoring Criteria of modified WCRF/AICR score

% of participants with maximum component scores

Components

Age Class (years)

Sex

0#

0.5Ç‚

<40

n=1322

41-49

n=1024

50-59

n=996

60-69

n=792

≥70

n=732

Female

n=2352

Male

n=2514

Total

n=4866

Energy dense foods (kcal/100g)

>175

125-175

≤125

30.83

27.14

28.84

23.63

18.61

33.16

20.64

26.70

Fast food intake (g/day)

>42

18-42

<18

1.25

1.61

5.61

14.30

20.69

7.70

6.76

7.21

Sugary drinks intake (g/day)

>250

≤250

0

6.78

14.53

24.73

36.53

44.44

23.04

21.93

22.47

Fruits and Vegetables (g)

<200

200-400

≥400

63.20

71.44

71.16

73.18

74.72

69.77

69.93

69.85

Cereals. Whole grain bread and Legumes

(g)

<24

24-64

≥64

27.21

19.88

20.57

12.64

10.83

20.66

18.54

19.56

White bread. pasta and rice (g/day)

≥144

91-144

<91

39.68

43.69

54.92

57.60

58.75

59.86

39.58

49.38

*Red (R) and processed meat (P)

R+P≥500 or P≥50

R+P<500 and P 3-50

R+P<500 and P<3

13.50

18.26

24.41

37.55

47.22

29.21

22.24

25.61

**Cold meat (g/day)

>22

7-22

≤7

14.97

19.98

31.79

48.02

59.03

35.25

27.68

31.34

Alcohol intake (g/day)

≥20

10-20

≤10

73.01

62.97

54.23

53.64

60.56

80.23

45.11

62.08

Sodium (g/day)

≥3

2.4-3

<2.4

62.68

71.36

79.97

84.36

88.34

79.17

71.04

74.98

***BMI (kg/m2)

<18.5

and≥30

25-30

18.5-24.9

52.80

36.73

23.33

16.86

18.33

39.33

26.09

32.49

  • Maximum score 1 was assigned when the recommendations were fully satisfied; Ç‚Intermediate score 0.5 when the recommendations were partially satisfied; #Low score 0 when the recommendations were partially satisfied

  • Lines 248-250 “High adherence scorers were ≥60 years old (about 40%) and women (73%). In contrast, Low Adherence scorers were younger (<60 years old, about 80%) and male (81%). High Adherence scorers were also married (72.7%), had Secondary School or higher Education (66.3%) and pensioners (31.5%).“  I suggest to reformulate text as to indicate that some characteristics were more frequent or dominated in e.g. in the group of people with the highest score. In present form it is slightly misleading despite percentages in brackets.

We have added a narrative description of participants. Now it reads:

Higher scorers were more likely to be older and women; they were also more likely to be married, with Secondary or higher educational level and pensioners.

  • Titles of supplementary materials – shouldn’t be Sensitivity instead of Sensibility, I'm not sure whether in a statistic both these terms are used interchangeably.

We have changed sensibility for sensitivity as they are different meanings.

This manuscript is a resubmission of an earlier submission. The following is a list of the peer review reports and author responses from that submission.

Round 1

Reviewer 1 Report

This paper focused on the evaluation of the use of a scoring system in real life, with the final aim to depict the risk of mortality associated with scored lifestyles.

This is a nicely written a well-conducted study. I have just a remark to add regarding the effect of age. Since obviously mortality increases with age, I believe that Figure 1 is not fully informative. I would recommend the authors to show not the absolute effect of age but the relative effect, as compared with the expected lifespan according to age and gender. This is referred to the work of Ederer first and then Hakulinen. In this way, the impact of a poor lifestyle will be expressed in terms of loss of potential life-years.

Minor:

  1. the authors stated that the cohort was overlapping in distribution with the general population. they should add data to inform the reader about
  2. Line 407 and 409 replace sensibility with sensitivity

Author Response

Answers to Reviewer 1

This paper focused on the evaluation of the use of a scoring system in real life, with the final aim to depict the risk of mortality associated with scored lifestyles.

  • This is a nicely written a well-conducted study. I have just a remark to add regarding the effect of age. Since obviously mortality increases with age, I believe that Figure 1 is not fully informative. I would recommend the authors to show not the absolute effect of age but the relative effect, as compared with the expected lifespan according to age and gender. This is referred to the work of Ederer first and then Hakulinen. In this way, the impact of a poor lifestyle will be expressed in terms of loss of potential life-years.

We thank the comment by the reviewer. Indeed, we had considered the use of relative survival statistical techniques. However, our cohorts are small and we observed that the above mentioned statistical technique has been mostly used with national cancer registries. By applying relative survival analysis we would have obtained imprecise and highly unstable estimates. To partially overcome this problem we choose age at death as time axis rather than time from birth. Moreover, in our region life expectancy is higher than national life expectancy.

Minor:

  1. the authors stated that the cohort was overlapping in distribution with the general population. they should add data to inform the reader about

We thank the reviewer. We have changed the paragraph. Now it reads: The Nutrition Hepatology (NUTRIHEP) Study is a cohort of subjects enrolled in 2005-06 from the city of Putignano (Apulia, Southern Italy)Using a systematic random 1-in-5 sampling procedure, a sample of the general population > 18 years of age was drawn from the General Practioner ’ s list of records. We used the records of GP, instead of a drawing from the census, because no significant difference was found between the distribution of the general population from Putignano and the subjects inscribed in GPs ’ records. In Italy, it is stated by law that everybody should have a GP. Therefore, the list of general population in the GP offices corresponds to the census list. A possible selection bias lies in the fact that specific sub-cohorts of patients (i.e., senior patients, patients with chronic diseases, patients with known chronic liver disease) would be more likely to be seen by the GP over a limited period of time. To minimize this potential confounding, a statistical analysis was carried out to test whether the mean age of the general population was comparable with that of subjects recruited by GP clinics. Therefore, we used one-way analysis of variance (ANOVA) and Bonferroni’ s test for multiple comparisons. ANOVA was then used to test the hypothesis that sex-specific mean age was the same among the general population and subjects of the GP clinics. There was no statistical evidence on differences in the mean age (p= 0.15). [19]

  1. Line 407 and 409 replace sensibility with sensitivity

We have changed sensibility for sensitivity

Reviewer 2 Report

The manuscript examined the association between modified WCRF/AICR score and all cause and cause specific mortality in a southern Italian population. The study reported a 41% lower risk among individuals with the highest mWCRF/AICR score compared with those who had the lowest. While the topic is interesting, I will discuss a few concerns.

Introduction

The manuscript began with rather broad summary of diet and health, which did not address the current literature on the relationship between WCRF/AICR score and mortality sufficiently. I do not understand why this paper is necessary after reading the introduction given many existing studies have reported significant associations between adherence to WCRF/AICR recommendation and mortality. It is not clear to me how this study fills a gap in the literature.

The authors mentioned Southern Italy and Mediterranean diet but did not elaborate why this geographic region and diet pattern may be an important factor in understanding the relationship between diet quality and mortality. Moreover, the manuscript consistently stratified the analyses by sex, and yet provided no rationale as why this demographic characteristic could be a critical moderator of the relationship.  

The manuscript stressed on competing risk analyses, but no justification was discussed. Why is it necessary? How does it compare with other approaches?

Methods

The description of the two cohorts is inadequate. Supplementary Figure 1 should illustrate the sample selection by cohort separately. Retention rates were not provided. Also, the supplementary Figure 1 showed that 642/5913 refused to participate, how did that translate to a 98.0% response rate (the citated study by Osella et al. reported a response rate of 70.6% for the same cohort). The authors mentioned excluding 366 subjects and yet the supplementary figure indicated that 405 subjects were excluded due to missing data. The inconsistency is very confusing.

Table 2 showed that close to 20% subjects had missing data on diabetes, cancer, dyslipidemia and hypertension, how were the missing data in the covariates handled?

The exposure assessment covered only 5 of the 10 recommendations, why other dimensions of the WCRF/AICR not assessed?

The description of the measurement is insufficient, and there appears to be inconsistency again. For example, the authors described the score consisted of 11 items and yet Table 1 had only 10. Moreover, the items were supposed to be linked with objectives including “maintain adequate body weight”, but none of the include items measures this objective!

More details should be provided on the selection of cut-off point for each item and the grouping of the overall score. Also, the items listed in Table 1 are difficult to understand, especially for general readers. For example, what is “cold meat” and “Na”?

Was the mortality tracing complete (i.e. were vital status verified for all subjects)? What was the end date of mortality tracing?

Given the relatively large number of death due to “other cause” and the cancer-rated aim of WCRF/AICR score, why cancer mortality was not estimated?

If ANOVA and Chi-squared tests were performed to test differences, where were the results reported? At least, table 2 included no such results.

Why were education, job type, diabetes, and cancer not adjusted for? Again, how were missing data handled with covariates? Given the strong predictive power of cigarette smoking, why merge never and former smoking in a single group?

What was the rationale of using AIC/BIC to select age range at death? Shouldn’t age range determined a prior, e.g., predefining a age range for premature mortality?

Results

If the purpose was to compare baseline characteristics among groups with different WCRF/AICR score, why not consistently report column percentages in Table 2? The authors reported that “Among participants, the highest scores were obtained in females (76% versus 23.9% of males), whereas among the lowest scorers, men accounted for 68.5% versus 31.4% of females” – weren’t these numbers column percentages?! 

Please report the number of death and mortality rates along with HRs.

What is “IC95%”?! I understand it should be “95% CI” but this type of error can be easily avoided by careful proofreading. Language problems were evident in many other places. For example, lines 158-161, the variable names should not be capitalized. Lines 190-191 should not be a separated paragraph and the sentence does not have a verb!

Discussion

Please be refrained in using the word “effect” (line 281). This observational study alone cannot establish causality.

The authors should be more specific as how the results compare with those from other studies, and how the present study extended the current literature.

The manuscript did not assess mortality risk associated with components of the WCRF/AICR score, and yet 4.2 focused on low-fiber diet and health?! Moreover, 4.3 diet rich in vegetable foods and its impact on the environment is not relevant! The analyses had nothing to do with food and environment.

The discussion section provided no clear speculation/theory as why adherence to WCRF/AICR benefit males more than female.   

Author Response

Answer to Reviewer 2

The manuscript examined the association between modified WCRF/AICR score and all cause and cause specific mortality in a southern Italian population. The study reported a 41% lower risk among individuals with the highest mWCRF/AICR score compared with those who had the lowest. While the topic is interesting, I will discuss a few concerns.

Introduction

  • The manuscript began with rather broad summary of diet and health, which did not address the current literature on the relationship between WCRF/AICR score and mortality sufficiently. I do not understand why this paper is necessary after reading the introduction given many existing studies have reported significant associations between adherence to WCRF/AICR recommendation and mortality. It is not clear to me how this study fills a gap in the literature.
  • The authors mentioned Southern Italy and Mediterranean diet but did not elaborate why this geographic region and diet pattern may be an important factor in understanding the relationship between diet quality and mortality.
  • Moreover, the manuscript consistently stratified the analyses by sex, and yet provided no rationale as why this demographic characteristic could be a critical moderator of the relationship.  
  • The manuscript stressed on competing risk analyses, but no justification was discussed. Why is it necessary? How does it compare with other approaches?

We thank the reviewer. We have completely changed the introduction section: Now it reads:

Introduction

A decreased morbidity and improvements in the desired quality of life can be achieved in a population by means of health promotion, when this takes deep root in the consciousness of that population [1]. Scientific efforts to elucidate the relationship between nutrition and health have greatly improved our understanding of the association between diet and health. Nutritional conditions in real life, in healthy individuals who have an adequate diet, do not depend only on individual ingredients or products, but also on a correct understanding of the idea of a "balanced diet", since the human metabolism features a great capacity for flexibility [2]. The relevance of nutrition science lies primarily in the growing knowledge of the long-term impact of nutrients, foods, and eating patterns on both health maintenance and disease onset [3].This requires studies to be expanded to adjacent scientific fields beyond biomedical domains, such as social sciences and data sciences, in order to better understand what drives humans to desire the foods they eat. Nutrition sciences are not only about the biochemical aspects, but also include cultural and behavioral elements, as well as environmental sustainability issues[4].

Scientific challenges promote the development of transdisciplinary disciplines and open science data (FAIR: Findable, Accessible, Interoperable and Reusable) [5].

Humans are currently facing a global transition in food production, [6] future breakthroughs in nutritional science will be strategic.The indications of the World Cancer Research Found are a reference not only for a correct diet, but also for physical activity, in the prevention of oncological diseases [7]. These indications are considered also a valid prevention tool for chronic diseases with risk factors related to eating habits[8-10]; these indications have been used in several observational studies in different populations [11-14].

The association of WCRF/AIRC with cancer, all-causes and cause-specific mortality has been extensively studied in several geographical areas. [9,15-17] This association has been explored with a variety of sites including colon, breast and pancreas cancers. [17]  WCRF/AIRC and lower all-cause mortality rate is the most prevalent association documented in literature[15] whereas not always an association has been found with cancer cause-specific mortality. [16] The beneficial effects of adherence to one or more WCRF/AIRC components have been also observed between recent and long-term cancer survivors. [17] Besides, a comparative study of six dietary indexes conducted in Iran did not find any association between WCRF/AIRC and cancer mortality[9] as well as a recent study from an area similar to ours between Mediterranean diet and cancer mortality. [18]

It is interesting to note that most studies aimed at probing the association between WCRF/AIRC and mortality relied on Cox’s survival model for all the associations considered including cause-specific mortality. [9,15,18] Cause-specific mortality is typical example of competing risks that frequently occur in epidemiologic studies but are often not recognized or ignored. [19,20] The use of classical survival analysis to estimate the incidence function and sub-distribution hazards ratios (SHR) may result in upward biased estimates. [21] Then, an appropriate statistical methodology should be applied. [22,23]

Mediterranean diet is the most prevalent dietary pattern in this geographical area and their dietary omponents as well other recommendations included recently in the Mediterranean diet pyramid, [24] fit very close to WCRF/AIRC recommendations. [25] Furthermore, the adherence to Mediterranean diet in this area seems to have changed little over time with a differential adherence between sexes. [26]

Our Institution, the National Institute of Gastroenterology ‘S de Bellis’ Research Hospital has conducted several epidemiological studies in this area and has documented a negative high age-related prevalence and a low incidence of Hepatitis C Virus infection. This infection produces a wide spectrum of gastrointestinal diseases from simple hepatitis to Hepatocellular Carcinoma. So, we thought that considering the Digestive Diseases-related (DSD) deaths rather than only Digestive System cancers, could more faithfully represent the cause-specific mortality in our study. This approach is further reinforced by the fact that only 19.4 % and 7.2%of deaths were cancers and Digestive System cancers, respectively. [27]

 For this purpose we built up a modified WCRF/AICR score (mWCRF/AICR score), following the WCRF/AICR indications, introducing some changes related to our study population [7,11,28-30].

This prospective cohort study conducted in Southern Italy during the period 2005-2017 was aimed at estimating the effect of adherence to a mWCRF/AICR score on DSD-related mortality, Cardiovascular Diseases (CVD)-related mortality and Other-Cause mortality (OCD). The score is intended as a tool for investigating a balanced and healthy diet, from the perspective described above, which can be readily used as an investigation tool also in clinical practice, in order to promote dietary and lifestyle behaviors aimed at maintaining a state of good health.

Methods

  • The description of the two cohorts is inadequate. Supplementary Figure 1 should illustrate the sample selection by cohort separately. Retention rates were not provided. Also, the supplementary Figure 1 showed that 642/5913 refused to participate, how did that translate to a 98.0% response rate (the citated study by Osella et al. reported a response rate of 70.6% for the same cohort). The authors mentioned excluding 366 subjects and yet the supplementary figure indicated that 405 subjects were excluded due to missing data. The inconsistency is very confusing.

We thank the reviewer. The text was misleading as it was not very clear that our baseline was fixed at 2005-2006. Response rate 70.6%  is referred to the response rate for the MICOL enrollment during 1985-86.

We have now included two flowcharts (one for each study) and changed the text. The texts now reads: “Considering as the baseline for both studies a total of 6114 subjects were invited to participate.

For the Micol / Panel cohort a total of 3614 subjects were invited to participate (of which 1708 Micol study and 1906 Panel study). Of these, 2970 (82.2% response rate) accepted to participate.We excluded 122 subjects for incomplete information (110 missing Food frequency Questionnaire and 12 missing Body Mass Index measurements). Finally, 2848 subjects (78.8% inclusion rate) were included in the study.

For the the NUTRIHEP cohort 2500 persons were invited to participate and 2301 accepted (92% response rate). We excluded 283 subjects for incomplete information (256 missing Food Frequency Questionnaire and 27 missing Body Mass Index measurements), so 2118 subjects (80.7% inclusion rate) were included in the study

Therefore, 5271  out of 6114 (86.2% response rate)  accepted to participate and 4866 out of 6114 (79.6% inclusion rate) subjects were finally included.

  • Table 2 showed that close to 20% subjects had missing data on diabetes, cancer, dyslipidemia and hypertension, how were the missing data in the covariates handled?

We have modified the text in the statistical analysis section. Now it reads:

Variables which had not available information such as diabetes, cancer, dyslipidemia and hypertension, the missing values were handled as a third category in the variable. In the modelling step a likelihood ratio test was performed to include or not each variable.

  • and 4) The exposure assessment covered only 5 of the 10 recommendations, why other dimensions of the WCRF/AICR not assessed?

The description of the measurement is insufficient, and there appears to be inconsistency again. For example, the authors described the score consisted of 11 items and yet Table 1 had only 10.

We have changed Table 1 as BMI was missing. Furthernore we have modified the text. Now, it redas:

This has now been included in the statistical analysis section. The text now reads: “Adherence to the WCRF/AICR indications was estimated with the WCRF/AICR score 12-14.The scoring system was built up referring to the WCRF/AICR indications applied to EpiGEICAM data 12. The score consisted of 11 items linked to the following domains: a) maintain adequate body weight, b) limit the consumption of energy-dense foods, c) eat mostly foods of plant origin, d) limit the intake of red meat and avoid processed meat, e) limit alcoholic drinks and the consumption of salt and salt-preserved foods. For each of 11 items a maximum score of 1 was assigned when the recommendations were fully satisfied, a value of 0 when the recommendations were not satisfied and 0.5 points as an intermediate score. Higher scores indicate a greater concordance with the WCRF/AICR recommendations 12,13.

We were not able to evaluate weight history, physical activity and dietary supplement use because these informations were not available.

  • More details should be provided on the selection of cut-off point for each item and the grouping of the overall score. Also, the items listed in Table 1 are difficult to understand, especially for general readers. For example, what is “cold meat” and “Na”?

We have modified the text, included it in the statistical analysis section and added a reference. The text now reads: “To assign the score to each of 11 items assessed cut-off points were taken from literature. 12

A legend has been added as a footnote in table 1 to specify what Cold meat and Na mean. The table now reads:

Scoring Criteria of modified WCRF/AICR score

% of partecipants with maximum component scores

Components

Age Class (years)

Sex

0

0.5

1

<40

n=1322

41-49

n=1024

50-59

n=996

60-69

n=792

≥70

n=732

Female

n=2352

Male

n=2514

Total

n=4866

Energy dense foods (kcal/100g)

>175

125-175

≤125

30.83

27.14

28.84

23.63

18.61

33.16

20.64

26.70

Fast food intake (g/day)

>42

18-42

<18

1.25

1.61

5.61

14.30

20.69

7.70

6.76

7.21

Sugary drinks intake (g/day)

>250

≤250

0

6.78

14.53

24.73

36.53

44.44

23.04

21.93

22.47

Fruits and Vegetables (g)

<200

200-400

≥400

63.20

71.44

71.16

73.18

74.72

69.77

69.93

69.85

Cereals. Whole grain bread and Legumes

(g)

<24

24-64

≥64

27.21

19.88

20.57

12.64

10.83

20.66

18.54

19.56

White bread. pasta and rice (g/day)

≥144

91-144

<91

39.68

43.69

54.92

57.60

58.75

59.86

39.58

49.38

Red (R) and processed meat (P)*

R+P≥500 or P≥50

R+P<500 and P 3-50

R+P<500 and P<3

13.50

18.26

24.41

37.55

47.22

29.21

22.24

25.61

Cold meat (g/day)**

>22

7-22

≤7

14.97

19.98

31.79

48.02

59.03

35.25

27.68

31.34

Alcohol intake (g/day)

≥20

10-20

≤10

73.01

62.97

54.23

53.64

60.56

80.23

45.11

62.08

Sodium (g/day)

≥3

2.4-3

<2.4

62.68

71.36

79.97

84.36

88.34

79.17

71.04

74.98

BMI (kg/m2)***

<18.5

25-30

18.5-24.9

52.80

36.73

23.33

16.86

18.33

39.33

26.09

32.49

*Red and Processed meat g/week; Processed meat g/day ; **Cold meat: meats subjected to a salting process; ***BMI: Body Mass Index

  • Was the mortality tracing complete (i.e. were vital status verified for all subjects)? What was the end date of mortality tracing?

We have added in the 2.3. Tracing Procedures and Outcome Assessment the following sentence: It was possible to trace the vital status for all included persons.

The time-to event is stated in the statical analysis section: “Time from enrollment to death, moving elsewhere or end of the study (December 31st, 2017), whichever occurred first.”

  • Given the relatively large number of death due to “other cause” and the cancer-rated aim of WCRF/AICR score, why cancer mortality was not estimated?

  • If ANOVA and Chi-squared tests were performed to test differences, where were the results reported? At least, table 2 included no such results.

We have added a new column in Table which contains p values for ANOVA and Chi-squared

The table now reads:

TABLE 2 Characteristics of Participants by modified WCRF/AICR Score Categories

MICOL/PANEL and NUTRIHEP Studies. Castellana Grotte. Putignano (BA). Italy. 2005-2017

All ¥

Modified WCRF/AICR score categories§

0-11

≤5

5.5-7

>7

p-value

N***

4866 (100.0)

2067 (42.5)

2194 (45.1)

605 (12.4)

Age at Enrollment (years)*

51.4 (15.8)

48.8 (14.9)

52.7 (16.05)

55.9 (16.3)

<0.001

Age (categorical, years)***

<40

1356 (27.8)

666 (49.1)

569 (42.0)

121 (8.9)

40-49

991 (20.4)

496 (50.1)

402 (40.6)

93 (9.4)

50-59

1016 (20.9)

420 (41.3)

463 (45.6)

133 (13.1)

60-69

783 (16.1)

267 (34.1)

395 (50.4)

121 (15.5)

≥70

720 (14.8)

218 (30.3)

365 (50.7)

137 (19.0)

<0.001

Sex***

Female

2352 (48.3)

650 (27.6)

1242 (52.8)

460 (19.6)

Male

2514 (51.7)

1417 (56.4)

952 (37.9)

145 (5.8)

<0.001

Marital Status***

Single

763 (15.7)

337 (44.2)

337 (44.2)

89 (11.7)

Married/Coupled

3661 (75.2)

1580 (43.2)

1635 (44.7)

446 (12.2)

Separated/Divorced

117 (2.4)

48 (41.0)

55 (47.0)

14 (12.0)

Widower

312 (6.4)

93 (29.8)

163 (52.2)

56 (17.9)

Without information

13 (0.3)

9 (69.2)

4 (30.8)

0 (0.0)

<0.001

Education***

Primary School

1329 (27.3)

493 (37.1)

645 (48.5)

191 (14.4)

Secondary School

1480 (30.4)

684 (46.2)

625 (42.2)

171 (11.6)

High School

1388 (28.5)

610 (43.9)

617 (44.5)

161 (11.6)

Graduated

493 (10.1)

223 (45.2)

217 (44.0)

53 (10.8)

Illiterate

169 (3.5)

55 (32.5)

85 (50.3)

29 (17.2)

Without Information

7 (0.1)

2 (28.6)

5 (71.4)

0 (0.0)

<0.001

Job***

Managers & Professionals

285 (5.9)

143 (50.2)

115 (40.4)

27 (9.5)

Craft. Agricultural and

Sales Workers

1176 (24.2)

613 (52.1)

476 (40.5)

87 (7.4)

Elementary Occupations

1127 (23.2)

510 (45.3)

494 (43.8)

123 (10.9)

Housewife

632 (13.0)

182 (28.8)

321 (50.8)

129 (20.4)

Pensioneers

1367 (28.1)

502 (36.7)

654 (47.8)

211 (15.4)

Jobless

253 (5.2)

105 (41.5)

121 (47.8)

27 (10.7)

Without Information

26 (0.5)

12 (46.2)

13 (50.0)

1 (3.8)

<0.001

DBP (mmHg)*

76.7 (9.6)

122.6 (17.4)

124.6 (18.1)

127.4 (17.5)

<0.001

SBP (mmHg)*

124.1 (17.8)

76.2 (9.6)

76.7 (9.7)

78.9 (9.4)

<0.001

Weight (kg)*

73.1 (14.9)

77.8 (15.5)

70.9 (13.8)

64.8 (11.2)

<0.001

BMI (kg/m2)*

27.5 (5.2)

28.5 (5.3)

27.1 (5.0)

25.6 (4.2)

<0.001

Kcal days

2173.5 (818.5)

2617.2 (848.2)

1942.2 (606.5)

1496.3 (519.9)

<0.001

Triglycerides (mmol/L)*

1.4 (1.0)

1.5 (1.1)

1.3 (0.9)

1.2 (0.8)

<0.001

Total Cholesterol (mmol/L)*

5.1 (1.0)

5.1 (1.0)

5.1 (0.9)

5.1 (1.09)

0.34

HDL (mmol/L)*

1.3 (0.3)

1.3 (0.3)

1.3 (0.4)

1.4 (0.4)

<0.001

LDL (mmol/L)*

3.1 (0.9)

3.2 (0.9)

3.1 (0.9)

3.1 (0.9)

0.038

RChol(mmol/L)*

0.6 (0.5)

0.7 (0.5)

0.6 (0.4)

0.6 (0.4)

<0.001

Glucose (mmol/L)*

5.9 (1.4)

5.9 (1.2)

5.9 (1.4)

5.8 (1.7)

0.39

ALT (μkat/L)*

16.7 (13.4)

0.3 (0.2)

0.3 (0.2)

0.2 (0.3)

<0.001

Smoke***

Never/Former

4024 (82.7)

1645 (40.9)

1832 (45.5)

547 (13.6)

Current

837 (17.2)

419 (50.1)

361 (43.1)

57 (6.8)

Without Information

5 (0.1)

3 (60.0)

1 (20.0)

1 (20.0)

<0.001

Observation time**

11.87 (11.2-12.2)

11.91 (11.5-12.2)

11.85 (11.2-12.2)

11.78 (11.2-11.9)

Status***

Alive and/or Censored

4381 (90.1)

1888 (43.1)

1954 (44.6)

540 (12.3)

Dead

484 (9.9)

179 (37.0)

240 (49.6)

65 (13.4)

0.036

Cause of Death***

Alive and/or Censored

4381 (90.0)

1887 (43.1)

1954 (44.6)

540 (12.3)

DSD-related mortality

82 (1.7)

29 (35.4)

43 (52.4)

10 (12.2)

CVD-related mortality

131 (2.7)

46 (35.1)

68 (51.9)

17 (13.0)

Other-Cause mortality

272 (5.6)

105 (38.6)

129 (47.4)

38 (14.0)

0.13

Diabetes***

No

3611 (74.2)

1676 (46.4)

1588 (44.0)

347 (9.6)

Yes

293 (6.0)

112 (38.2)

142 (48.5)

39 (13.3)

Without Information

962 (19.8)

279 (29.0)

464 (48.2)

219 (22.8)

0.011

Cancer diagnosis***

No

3740 (76.9)

1733 (46.3)

1646 (44.0)

361 (9.7)

Yes

158 (3.2)

55 (34.8)

78 (49.4)

25 (15.8)

Without Information

968 (19.9)

279 (28.8)

470 (48.6)

219 (22.6)

0.004

Dyslipidemia ***

No

3139 (64.5)

1433 (45.7)

1383 (44.1)

323 (10.3)

Yes

762 (15.7)

355 (46.6)

345 (45.3)

62 (8.1)

Without Information

965 (19.8)

279 (28.9)

466 (48.3)

220 (22.8)

0.20

Hypertension***

No

2761 (56.7)

1324 (48.0)

1177 (42.6)

260 (9.4)

Yes

1141 (23.4)

464 (40.7)

551 (48.3)

126 (11.0)

Without Information

964 (19.8)

279 (28.9)

466 (48.3)

219 (22.7)

<0.001

ALT: Alanine Aminotrasferase; BMI: Body Mass Index; DBP: Diastolic Blood Pressure; SBP: Systolic Blood Pressure; HDL: High Density Lipoprotein Cholesterol; LDL: Low Density Lipoprotein Cholesterol; DSD-related mortality: Digestive System Disease-related mortality;CVD-related mortality: Cardiovascular Disease-related mortality; RChol: Remnant Cholesterol. Cells reporting subject characteristics contain * Mean±(SD). **Median (IQR). ***Number. ¥ (Percentage) Percentages calculated per column. §(Percentage) Percentage calculated per row

  • Why were education, job type, diabetes, and cancer not adjusted for? Again, how were missing data handled with covariates? Given the strong predictive power of cigarette smoking, why merge never and former smoking in a single group?

The variables education, job type, diabetes, and cancer were not used as adjusting variables because of the likelihood ratio test was not significant. We preferred to save degrees of freedom as the estimates did not change.

We have change the text in the statistical analysis section. Now it reads: smoking never/former (quit smoking 5 years or more) vs current

  • What was the rationale of using AIC/BIC to select age range at death? Shouldn’t age range determined a prior, e.g., predefining a age range for premature mortality?

We thank the reviewer as the concept was not clear. We have changed the paragraph in the statistical analysis section. Now it reads:

Before fitting survival models, to overcome the problem of a few events in the tails of age at death distribution, we performed a separate analysis to establish the best range of age at death. In this analysis, we applied flexible parametric competing risk survival model to the data, using different ranges of age at death and mWCRF/AICR for the whole cohort and sex and by using Akaike's (AIC) and Schwarz's Bayesian Information (BIC) Criteria to select the best performing model.

Results

  • If the purpose was to compare baseline characteristics among groups with different WCRF/AICR score, why not consistently report column percentages in Table 2? The authors reported that “Among participants, the highest scores were obtained in females (76% versus 23.9% of males), whereas among the lowest scorers, men accounted for 68.5% versus 31.4% of females” – weren’t these numbers column percentages?! 

We have changed the text and considered only row percentages. The paragraph now reads:

The highest score was reached by 19.6% of females and 5.8% of males. There was an increasing trend of adherence with increased age class. Among widowers at least 17.9% achieved the highest score, illiterates until to 17.2% and among housewives until to 20.4%

  • Please report the number of death and mortality rates along with HRs.

We have now included a new table (Supplementary material) reporting number of deaths and mortality rates and modified the text. It now reads:

Number of deaths and rates with their corresponding 95%CI for the whole cohort, sex and all-cause and cause specific mortality are shown in Supplementary table 6.

  • What is “IC95%”?! I understand it should be “95% CI” but this type of error can be easily avoided by careful proofreading. Language problems were evident in many other places. For example, lines 158-161, the variable names should not be capitalized.

We have corrected these mistakes.

Lines 190-191 should not be a separated paragraph and the sentence does not have a verb!

We have changed the paragraph. It now reads:

All statistical analyses were performed by using Stata, statistical software version 16.1 (StataCorp, 4905 Lakeway Drive, College Station, Texas 77845 USA), in particular, the user-written-stpm2cr-routine and post-estimation commands (24).

Discussion

  • Please be refrained in using the word “effect” (line 281). This observational study alone cannot establish causality.
  • The authors should be more specific as how the results compare with those from other studies, and how the present study extended the current literature.
  • The manuscript did not assess mortality risk associated with components of the WCRF/AICR score, and yet 4.2 focused on low-fiber diet and health?! Moreover, 4.3 diet rich in vegetable foods and its impact on the environment is not relevant! The analyses had nothing to do with food and environment.
  • The discussion section provided no clear speculation/theory as why adherence to WCRF/AICR benefit males more than female.

We have changed the discussion following the reviewer suggestions. Now it reads:

  1. Discussion

In this cohort study conducted in southern Italy where the Mediterranean Diet is most prevalent way of eating, high adherence to mWCRF/AICR showed an important protective effect on all-cause mortality in the whole cohort and the male sub-cohort as well as in particular cause-specific mortality scenarios. DSD-related mortality resulted decreased when the adherence to mWCRF/AICR score was high  in the male sub-cohort as  CVD-related mortality was for the female sub-cohort. Instead, for Other-Cause mortality high adherence to mWCRF/AICR score resulted in a significant reduction of the risk of mortality in the whole sample and in the females sub-cohort.

The association between a healthy diet, with a reduced risk of all-causes mortality, and the incidence of major chronic diseases has been shown in large epidemiologic studies [40,41]. Estimated effects of the Mediterranean Diet such as reduction of all-causes mortality and cause-specific mortality [42,43], reduced incidence of cardiovascular and cerebrovascular diseases, reduced incidence of neoplastic diseases  and neurodegenerative diseases as well as other clinical outcomes, such as stroke and mild cognitive disorders have been  [44]. Furthermore, results of the SU.VI.MAX trial suggest that antioxidants may contribute to counteract some of the potential deleterious effects of a pro-inflammatory diet on mortality [45]. The Mediterranean style with a low inflammatory potential may be associated with better outcomes [46]. Non-communicable diseases (NCD) are the main cause of death in developed countries [47]. Changes in lifestyles, especially diet, could play an essential role in preventing NCD and premature mortality [48]. Dietary patterns associated with a lack of compliance with WCRF/AICR recommendations, have been associated with higher concentration of inflammatory markers [49], and a higher intake of fruit and vegetables is associated with a lower mortality. Indeed, in a recent study the risk reduction showed a plateau at ≈5 servings of fruit and vegetables per day. These findings support current dietary recommendations to increase the intake of fruits and vegetables[50]. In the EPIC and study, participants reporting the consumption of more than 569 g/day of fruits and vegetables, had lower risks of death from diseases of the circulatory, respiratory and digestive system, when compared with participants consuming less than 249 g/day. The lower risk of death associated with a higher consumption of fruits and vegetables may be the result of inverse associations with diseases of the circulatory, respiratory and digestive systems[51]. Emerging evidence has shown that lifestyle, including diet, after a Colorectal Cancer (CRC) diagnosis might affect all-cause and CRC specific mortality risk. In particular in terms of risk of relapse, mortality [52] and survival [53]. Other studies have highlighted how lifestyle and eating habits are associated with gastric adenocarcinoma [54].

Our results show that most of the participants with the highest total score had a high daily intake of fruit and vegetables. The findings are consistent with the consensus that plant-based diets are beneficial for health, as Burkitt had already hypothesized in 1969 [55]. Low fiber consumption in high-income countries could be linked to the high prevalence of western diseases in those populations[56]. Current recommendations for dietary fiber intake for adults in most European countries and in the US are between 30–35 g per day for males and between 25–32 g per day for females [57]. These findings were already confirmed in the EPIC cohort [58]. Fiber intake is correlated with the occurrence of cancer and diabetes, but also with all–causes and cause-specific cardiovascular mortality[59,60]. Dietary fiber is known to 1) improve laxation by increasing bulk and reducing the transit time of feces through the bowel; 2) increase the excretion of bile acids, estrogen, and fecal pro-carcinogens and carcinogens by binding to them; 3) lower serum cholesterol; 4) slow glucose absorption and improve insulin sensitivity; 5) lower blood pressure; 6) promote weight loss; 7) inhibit lipid peroxidation; and 8) have anti-inflammatory properties[59]. Moreover, mechanistic studies have shown that products of fiber fermentation in the colon could suppress colonic mucosal inflammation and carcinogens [61]. Short-chain fatty acids can affect the epigenome through metabolic regulatory receptors, and this can reduce obesity, diabetes, atherosclerosis, allergy, and cancer [62]. Moreover, studies conducted in different countries, highlighted strong associations between the consumption of ultra-processed foods and an increased risk of obesity and several other diet-related chronic diseases [63]. Increasing dietary fiber intake to 50 g/day is likely to increase the lifespan, improve the quality of life during the added years, and substantially reduce health-care costs. The recommendation to eat mostly foods of plant origin is included among the WCFR/AICR recommendations [7].

.

A more general adoption of plant-based diets could lead to benefits also for planetary health [64]. The Mediterranean diet is considered to be one of the environmentally friendly options [65]. The amount of animal-based foods in the diet, particularly meat and dairy products, is the most significant contributor to the harmful effects for the environment and to a suboptimal sustainability. [66]. As the diet influences not only our health but also the environment, dietary advice should take into account the environmental impact of the diet. An enhanced adherence to an eco-friendly diet like the Mediterranean diet is an important goal in society today [65].

Our study showed that for DSD-related mortality, only in the male sub-cohort emerged an association between a high score and a reduction in mortality risk. Gender differences have been reported also for dietary habits, as well as individual responses to dietary intake [67,68]. Indeed, there appears to be some resistance to following a healthy diet in males [69]. In fact, in our region adherence to a more healthy life-style including dietary habits is higher among females and this adherence has little changed over time. [26] Masculinities is a term now commonly used to denote diversity and complexity among males and forms of masculine identity; there are suggestions that conventional masculinities play a, negative role in male health [71]. In a Scandinavian study which specifically considered males, masculinities and food preferences, several professional groups were sampled and it was found that working class males, carpenters, interpreted food in terms of fuel and rejected the traditional associations between food and health. On the contrary, middle class participants ,professional engineers, saw food more in terms of pleasure and of enjoying good food and a good life [72].

Our study has several strengths, particularly the cohort design and the large random population sample from a geographic area where the Med Diet is widespread. Moreover, the complete exposure assessment was performed using a recognized score [12]. In a 1997 validity study conducted by the EPIC study group in Italy, they compared results of the FFQ with 24-hour recall diet assessment. Subsequent modifications were made to the questionnaire by the EPIC study team, expected to decrease measurement errors. Our study did not include a measure of physical activity, a potentially serious limitation given previous research findings linking physical activity with all-causes and cause-specific mortality. Thus, we may have over- or underestimated the effect of diet due to a confounding or effect modification of physical activity [73]. Our score refers to the 2007 WCRF/AICR recommendations for the prevention of cancer; compared to those issued in 2017, the recommendation to limit the consumption of salt was more specific.

  1. Conclusions

Primary prevention is the most effective and economical approach to prevent chronic diseases. Emphasizing the quality of the diet and quantity may be the best preventive measure to achieve long-term personal and social goals at every stage of life. The healthy life span loss is evident from the myriad of non-communicable diseases that are insidious and manifest with sudden cardiovascular events, liver failure, lung disease, diabetes and all types of tumors. Our study showed a reduction in the risk of mortality for the digestive system disease, in male subjects who had a high adherence to the score; and a reduction in the risk of mortality for the cardiovascular disease, in female subjects who had a high adherence to the score. And a reduction in mortality was observed for all other-causes mortality for whole sample and for females who had a high adherence to the score. Further in-depth studies would be useful to evaluate possible educational interventions to promote a healthy lifestyle in the vulnerable group of the population in the reference area

The novelty of this paper consists in our mWCRF/AICR score was applied to a Mediterranean population with an appropriate statistical methodology aimed to obtain valid and precise estimates. It is an easy-to-use tool in clinical practice that allows a simple evaluation of both the qualitative and quantitative aspects of the diet as well as complete life-style. It could also be used by patients themselves as an immediate self-assessment tool, aimed at fostering a greater awareness of their lifestyle habits. Last but not least, our score has a cross-border character and could be used to compare lifestyle habits in different populations.

Round 2

Reviewer 2 Report

The current revision represents an improvement over the initial submission. However, several issues remain conspicuous.  

First, the manuscript still has many errors and inconsistencies, which concerns me greatly. For example, Supplementary Table 2 shows that the number of deaths in MICOL/PANEL studies were 481, and Supplementary Table 3 shows that NUTRIHEP study has 125 deaths. How was it possible that the number of all-cause deaths in the combined sample of cohorts was 484? Moreover, Supplementary Table 6 shows that the total number of all-cause deaths in the combined sample was 418?!

In the response to report the number of deaths and mortality rates in Table 3, the authors opted to add a supplementary table, documenting the requested information. Reading the table, the number of CVD-related deaths were zero among female respondents who had a modified WCRF/AICR score of 5 or below (reference group). How could a relative risk be defined if the denominator is zero?!

Similarly, for DSD-related mortality only 1 death was recorded for female respondents who had a score of 5 or less. The sparse number of events could render the estimation unreliable. I don’t think it is feasible to stratify the analyses by sex.

The manuscript reported a protective association among female respondents with higher score (Table 3); however, in Supplementary Table 6, the mortality rates were 1.30, 1.81, and 2.26 among female respondents who had a WCRF/AICR score of <5, 5.5-7, and >7, respectively. These numbers would indicate a higher SHR among groups with higher scores.  

The authors responded that “the variables education, job type, diabetes, and cancer were not used as adjusting variables because of the likelihood ratio test was not significant. We preferred to save degrees of freedom as the estimates did not change.” This is very concerning because education and comorbidity (e.g., cancer and diabetes) should be strong predictors of mortality. If the likelihood ratio test was not significant, it may indicate an serious issue with the modeling strategy. Indeed, the test statistics in Table 3 indicate clear differences for these characteristics among the exposure groups, how could LR test not significant?

The authors did not address my question “given the relatively large number of death due to ‘other cause’ and the cancer-rated aim of WCRF/AICR score, why cancer mortality was not estimated?”   

Table 2 should report column percentages, so that it is easier to compare baseline characteristics of respondents with different WCRF/AICR score.

I have suggested that the English language should be improved; however, the manuscript still has many language errors. Some examples include lines 52-53, 75-77, 93-97, 122, and 368.